# JetBench: Benchmarking Vision Models for Jet Observables' Classification in Heavy-Ion Physics

## Abstract

Relativistic heavy-ion collisions provide a window into quark–gluon plasma formation, but extracting parameters such as the energy loss mechanism, strong coupling, $\alpha_s$, and virtuality scale, $Q_0$, has traditionally required costly Bayesian inference. We introduce **JetBench**, a benchmark for multi-parameter classification of heavy-ion events using the ANONYMIZED dataset. Each event is encoded as a $32 \times 32$ jet image with three targets: energy loss module, $\alpha_s$, and $Q_0$. We systematically evaluate CNNs (EfficientNetV2, ConvNeXt V2), Transformers (ViT-CoMer, Swin V2), and state space models (Mamba) under unified training. Results show saturated performance on energy loss ($\sim$100%), strong accuracy on $\alpha_s$ ($\sim$95%), and up to 78% on $Q_0$, with ViT-CoMer achieving the best joint accuracy (74.5%). Loss-weight ablations reveal trade-offs between tasks, with $Q_0$ emphasis improving recall at modest cost to $\alpha_s$. Probability calibration confirms errors follow physics continuity (e.g., $\alpha_s = 0.2/0.3$, $Q_0 = 2.0/2.5$). These findings establish JetBench as a scalable complement to Bayesian approaches. Code and preprocessing scripts are available at `ANONYMIZED url`.

## 1 Introduction

Relativistic heavy-ion collisions provide a unique window into the study of the quark–gluon plasma (QGP), a state of matter believed to have existed microseconds after the Big Bang. These high-energy nuclear collisions, studied at the Relativistic Heavy Ion Collider (RHIC) at Brookhaven National Laboratory and the Large Hadron Collider (LHC) at CERN, create extreme conditions under which quarks and gluons deconfine, enabling direct investigation of QGP properties Putschke et al. (2019); Kumar et al. (2020); Tachibana et al. (2024). A schematic overview of the collision stages and QGP evolution is provided in the Appendix A, Fig. 7 to illustrate how the final-state particle distributions measured at detectors emerge from the underlying medium dynamics.

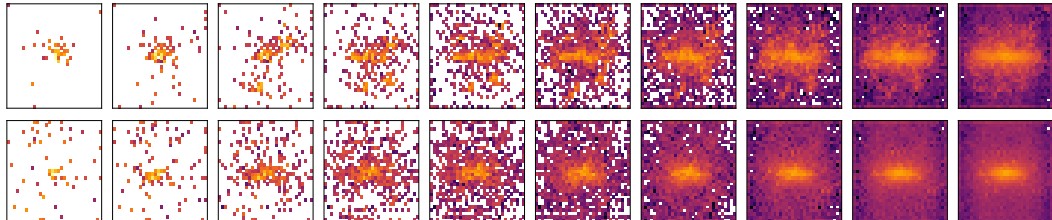

Figure 1: Evolution of averaged jet-event histograms for representative parameter settings. Each row shows the progressive aggregation of Pb–Pb collision events into averaged 2D histograms in $(\eta, \phi)$ space, with increasing sample counts ($n = 1, 2, 4, \ldots, 256, 500$). Top: $\alpha_s = 0.2$, $Q_0 = 1.0$, MATTER module. Bottom: $\alpha_s = 0.4$, $Q_0 = 2.5$, MATTER–LBT module.

A crucial observable in these collisions is the *jet* — a collimated spray of high-energy particles produced in hard scatterings. As these partons propagate through the QGP medium, they lose energy via multiple scattering and gluon radiation, a phenomenon known as *jet quenching*. Studying jets and

their modifications in the QGP allows physicists to probe the medium's properties and extract key parameters that govern parton–medium interactions. Figure 1 illustrates the evolution of averaged jet-event histograms in $(\eta, \phi)$ space for representative parameter settings, highlighting how individual sparse events converge into stable jet profiles as more events are aggregated. This representation corresponds to the detector-level final state of the schematic collision process (see Appendix A, Fig. 7) and forms the effective input to our ML-based parameter classification.

Building on this physics background, our ANONYMIZED dataset encodes each individual jet event as a $32 \times 32$ image in the pseudorapidity–azimuth $(\eta, \phi)$ plane. Each pixel stores the normalized transverse momentum $(p_T)$ content of all final-state particles falling into that bin, computed as the average summed $p_T$ over the particles in the cell and scaled by a global normalization constant. This representation transforms sparse detector-level information into a dense image format compatible with standard vision architectures. Figure 2 shows representative examples of such jet-event images, spanning values of the strong coupling constant $\alpha_s$, the virtuality separation scale $Q_0$, and different energy loss modules (MATTER or MATTER–LBT). These sparse images form the basis for our machine learning benchmark.

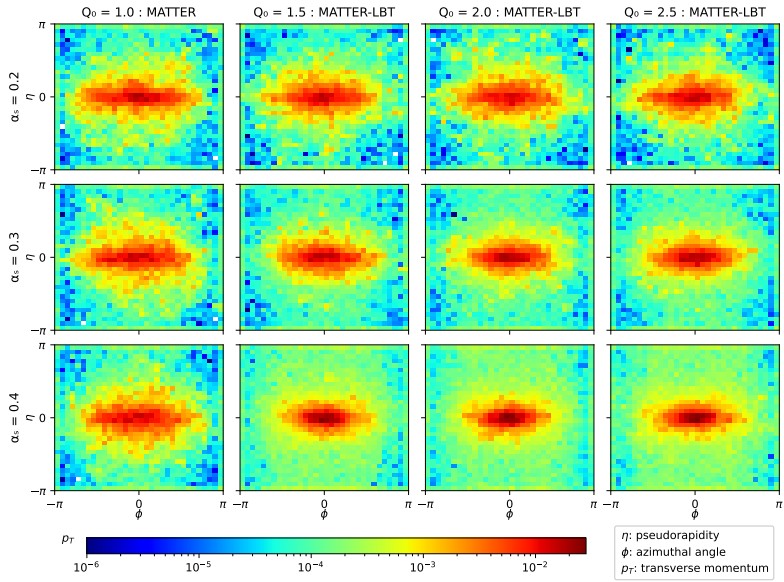

Figure 2: Representative grid of Pb–Pb collision events from the ANONYMIZED dataset. Columns vary by $\alpha_s \in \{0.2, 0.3, 0.4\}$, rows by $Q_0 \in \{1.0, 1.5, 2.0, 2.5\}$, and are labeled by the energy loss module (MATTER or MATTER–LBT). Each image is a $32 \times 32$ histogram in $(\eta, \phi)$ space, where the pixel intensity corresponds to the normalized sum of transverse momentum $p_T$ of all particles falling into that bin.

Modeling jet evolution requires complex multi-stage simulation frameworks such as JETSCAPE Putschke et al. (2019); Kumar et al. (2020), which couple parton transport modules (MATTER, LBT) with medium evolution. These simulations are controlled by key hyperparameters—notably $\alpha_s$ and $Q_0$—that must be tuned to match experimental data. Bayesian parameter estimation has been the gold standard Bernhard et al. (2016), but is computationally expensive, often requiring tens of thousands of full simulations. This has motivated use of deep learning as a scalable alternative, and recent work shows promise in emulating hydrodynamics Stewart & Putschke (2025) and anomaly detection in jet events ANONYMIZED (2025).

In our prior work ANONYMIZED (2025), we introduced ANONYMIZED, a benchmark dataset of 10.8M jet-event images, and demonstrated feasibility of classifying the energy loss module with CNNs. However, that study was limited to a single parameter. Here, we extend the problem to **multi-parameter classification**, jointly predicting $\alpha_s$, $Q_0$, and the energy loss module from event images. We benchmark modern vision architectures across paradigms—CNNs (EfficientNetV2, ConvNeXt V2), Vision Transformers (ViT-CoMer, Swin V2), and state space models (Mamba). Our results show that Vision Transformers achieve the strongest overall performance, particularly on $\alpha_s$

and $Q_0$ classification. Importantly, we go beyond accuracy: by analyzing the averaged probability distributions predicted by the models. We show that results are consistent with physics expectations, confirming that deep learning captures meaningful structure in heavy-ion collision data.

**Key Contributions.** (1) Extending the ANONYMIZED dataset analysis to **multi-parameter classification**, predicting $\alpha_s$, $Q_0$, and the energy loss module simultaneously. (2) Benchmarking state-of-the-art vision backbones—CNNs (EfficientNetV2, ConvNeXt V2), Vision Transformers (ViT-CoMer, Swin V2), and state space models (Mamba)—on scientific imagery of heavy-ion collisions. (3) Proposing a **moment-based aggregation framework** for representing jet events. We employ the *first moment* (mean $p_T$ per $(\eta, \phi)$ bin) to build stable image profiles from sparse events, showing that this preserves key physics characteristics. (4) Introducing a **physics-informed evaluation** using predicted probability distributions, demonstrating that models not only achieve high accuracy but also align with theoretical expectations from QCD-based energy loss. (5) Establishing ANONYMIZED as a vision benchmark that bridges computer vision and nuclear physics, positioning deep learning as a scalable complement to Bayesian analysis and continuing the trend of deep learning in ultra-relativistic heavy-ion collisions Stewart & Putschke (2025).

## 2 RELATED WORK

**Physics-Informed Machine Learning in Heavy-Ion Collisions.** Relativistic heavy-ion collisions generate extreme conditions where jets interact with the quark–gluon plasma (QGP), producing complex signatures that depend on parameters such as the energy loss module, strong coupling constant, $\alpha_s$, and virtuality separation scale, $Q_0$ Putschke et al. (2019); Kumar et al. (2020). Traditionally, Bayesian analysis has been the primary tool for parameter extraction Bernhard et al. (2016); Ehlers et al. (2022); Kumar et al. (2023); Tachibana et al. (2024), and continues to evolve with recent advances in calibration metrics Fan et al. (2024), multi-system modeling Mankolli (2024), and updated suppression analyses Ehlers et al. (2025). Although powerful, Bayesian inference remains computationally expensive, often requiring tens of thousands of simulations.

Recent physics studies underscore the complexity of jet-medium interactions. New results have examined photon-triggered jets Sirimanna et al. (2025b;a), hard jet substructure in multistage approaches Tachibana et al. (2024; 2025), hadronic reinteractions Roch et al. (2025), and in-medium hadronization Sengupta et al. (2025). These works highlight the richness of the QGP dynamics but also reinforce the need for scalable data-driven methods. Machine learning has begun to fill this gap: Stewart and Putschke Stewart & Putschke (2025) introduced Fourier Neural Operators to accelerate hydrodynamic evolution, pointing toward ML as a viable complement to Bayesian pipelines. Our work follows this trajectory by applying state-of-the-art vision models directly to jet-event images produced with JETSCAPE Putschke et al. (2019).

**Efficient CNN Architectures.** Despite the rise of sequence models, convolutional neural networks remain strong baselines, particularly for low-resolution images. EfficientNetV2 Tan & Le (2021) and ConvNeXt V2 Woo et al. (2023) represent the latest generation of CNNs, incorporating Transformer-inspired design elements such as depth scaling and improved normalization. Their efficiency and robustness make them well-suited for $32 \times 32$ jet-event images, allowing us to evaluate the gains of more complex architectures against competitive CNN baselines.

**Vision Transformers.** Vision Transformers (ViTs) have redefined image classification by modeling long-range dependencies Dosovitskiy et al. (2020). However, their lack of locality modeling has motivated hybrid approaches. ViT-CoMer Xia et al. (2024) introduces convolutional multiscale feature interactions into ViT backbones, enhancing local-global feature fusion. Hierarchical designs such as Swin Transformer V2 Liu et al. (2022) further improve efficiency via windowed attention and multiscale feature maps. These models are well aligned with jet-event images, which combine sparse localized deposits with broader global patterns.

**State Space Models and Mamba.** State space models present a recent alternative to self-attention for sequence learning. Mamba Gu & Dao (2023) leverages selective state spaces for linear-time modeling of long-range dependencies. Vision Mamba (Vim) Zhu et al. (2024) adapts this to vision tasks, demonstrating that state-space dynamics can replace attention while maintaining high performance. For jet-event classification, where localized fluctuations coexist with global structures, these models offer a computationally efficient and expressive solution.

Table 1: Model zoo summary for backbones evaluated on the ANONYMIZED dataset. We report the number of parameters, floating point operations (FLOPs), and average training time per model.

| Model | Params (M) | FLOPs (G) | Time (h) |
|---|---|---|---|
| EfficientNet V2 | 4.02 | 0.009 | 99.3 |
| ConvNeXt V2 | 27.83 | 0.091 | 72.3 |
| ViT-CoMer | 5.53 | 1.080 | 96.5 |
| Swin Transformer V2 | 27.50 | 0.090 | 73.5 |
| Mamba | 98.61 | 19.190 | 269.5 |

**Summary and Motivation.** Previous work on ANONYMIZED ANONYMIZED (2025) focused on binary classification of the energy loss module using VGG16 and point-based models. In contrast, we conduct the first systematic benchmark of modern CNNs, Vision Transformers, and state space models for *multi-parameter classification*, predicting $(E, \alpha_s, Q_0)$ jointly. By bridging recent advances in vision architectures with updated physics research on jet-medium dynamics, our study positions ANONYMIZED as both a vision benchmark and a scalable alternative to Bayesian analysis for parameter extraction in heavy-ion collisions.

## 3 METHODOLOGY

### 3.1 DATASET DESCRIPTION

We utilize a curated subset of the ANONYMIZED dataset introduced in ANONYMIZED (2025), focusing on 12 balanced combinations of physics parameters for structured multi-parameter classification. The dataset contains event images generated from Pb–Pb heavy-ion collision simulations using the JETSCAPE framework Putschke et al. (2019); Kumar et al. (2020). Each event is represented as a $32 \times 32$ pixel image encoding the transverse momentum ($p_T$) distribution of particles within the azimuthal–pseudorapidity ($\phi, \eta$) plane.

Each event is annotated with three key physics parameters: (1) the energy loss module, $E$, (2) the strong coupling constant, $\alpha_s$, and (3) the virtuality separation scale, $Q_0$. The formulation of these parameters within JETSCAPE simulations, including how $\alpha_s$ controls parton–medium interaction strength and how $Q_0$ sets the boundary between vacuum-like and medium-induced radiation, is detailed in our previous dataset paper ANONYMIZED (2025). Intuitively, $\alpha_s$ governs the strength of QCD interactions in the quark–gluon plasma, while $Q_0$ determines the scale at which jets transition from perturbative showers to medium-modified cascades. Formally, the dataset can be expressed as

$$\mathcal{D} = \{(x_i, y_i) \mid x_i \in \mathbb{R}^{32 \times 32}, \ y_i = (E_i, \alpha_{s,i}, Q_{0,i})\}, \tag{1}$$

where the label space is

$$\begin{aligned} E &\in \{\text{MATTER}, \text{MATTER-LBT}\}, \\ \alpha_s &\in \{0.2, 0.3, 0.4\}, \\ Q_0 &\in \{1.0, 1.5, 2.0, 2.5\}. \end{aligned} \tag{2}$$

Briefly, when $Q_0 = 1.0$, the module is always MATTER, yielding three $(E, \alpha_s, Q_0)$ combinations. For $Q_0 \in \{1.5, 2.0, 2.5\}$, the module is always MATTER–LBT, paired with the three $\alpha_s$ values, yielding nine additional combinations. Each combination contributes 600K event images (total 7.2M). This balanced construction induces a uniform label prior over the 12 valid 3-tuples,

$$P(E, \alpha_s, Q_0) = \tfrac{1}{12}, \quad \forall (E, \alpha_s, Q_0) \in \mathcal{Y}, \tag{3}$$

ensuring fair training and unbiased evaluation.

### 3.2 PROBLEM FORMULATION

Our goal is to train multi-output classifiers that jointly predict all three physics parameters from a given event image. Formally, given an input image $x \in \mathbb{R}^{32 \times 32}$, the model outputs $(y_1, y_2, y_3)$, where $y_1$ indicates $E$ (binary), $y_2$ indicates $\alpha_s$ (three-class), and $y_3$ indicates $Q_0$ (four-class). The training objective is a composite loss function:

$$\mathcal{L}_{\text{total}} = \lambda_E \mathcal{L}_{\text{binary}}(y_1, \hat{y}_1) + \lambda_{\alpha_s} \mathcal{L}_{\text{multi}}(y_2, \hat{y}_2) + \lambda_{Q_0} \mathcal{L}_{\text{multi}}(y_3, \hat{y}_3), \tag{4}$$

where $\lambda_i$ are task weights and $\hat{y}_i$ are the model predictions.

We evaluate a set of task-weighting schemes $(\lambda_{\mathrm{E}}, \lambda_{\alpha_s}, \lambda_{Q_0})$, defined in the Appendix D, Table 4 and referenced throughout our ablations.

**Task-Weighting Schemes.** Beyond the balanced baseline $(1, 1, 1)$, we design a small set of interpretable schemes (S1–S10) to probe cross-task trade-offs. Because $Q_0$ is the hardest head (4-way) and empirically drives most errors, several schemes tilt toward $Q_0$ (S2–S5, S9–S10). We also include mild/strong emphasis on $\alpha_s$ (S6–S7) and a modest "energy bump" (S8). The exact definitions of all schemes are provided in Appendix D, Table 4, and are referenced throughout our ablation tables (Section 4).

### 3.3 MOMENTUM-BASED AGGREGATION FOR STABLE JET PROFILES

Individual jet events are inherently sparse: most $(\eta, \phi)$ bins contain no particles, while a few bins carry large $p_T$ deposits ANONYMIZED (2025). This sparsity makes single-event images noisy and unstable for learning. To address this, we introduce the *Virtual Image Aggregation Algorithm*, which constructs stable jet profiles by combining multiple events with identical ground-truth labels into a single representative image.

Our procedure is motivated by statistical moments of the transverse momentum distribution. For event $e_k$ and pixel $(i, j)$, let $p_T^{(m,k)}$ denote the transverse momentum of particle $m$ in that cell, and $N_{ij}^{(k)}$ the particle count. The $r$th raw moment is defined as:

$$m_r^{(k)}(i,j) = \begin{cases} \frac{1}{N_{ij}^{(k)}} \sum_{m \in \text{cell}(i,j)} \left(p_T^{(m,k)}\right)^r, & N_{ij}^{(k)} > 0 \\ 0, & N_{ij}^{(k)} = 0, \end{cases} \tag{5}$$

and the aggregated moment profile across $n$ events is:

$$\bar{m}_r(i,j) = \frac{1}{n} \sum_{k=1}^{n} m_r^{(k)}(i,j), \quad I_{ij}^{(r)} = \frac{\bar{m}_r(i,j)}{\max_{i,j} \bar{m}_r(i,j)}. \tag{6}$$

In this work, we employ the **first moment** ($r = 1$), corresponding to the mean $p_T$ per $(\eta, \phi)$ bin. The resulting image $I^{(1)} \in [0,1]^{32 \times 32}$ encodes the normalized mean-$p_T$ field, which is both stable and physics-informed. In preliminary experiments, training on single-event images ($n = 1$) proved too noisy and failed to converge to meaningful results. We therefore constructed aggregated datasets with $n \in \{100, 500, 1000\}$ events per image. Among these, $n = 500$ yielded the most stable convergence across backbones by balancing noise reduction with sufficient sample diversity and was adopted as the default setting for all reported experiments. Figure 1 illustrates how aggregation transforms sparse single events into reproducible jet profiles as the sample count grows ($n = 1, 2, 4, \ldots, 500$). A large-scale visualization across all parameter combinations is provided in the Appendix C, Fig. 8. A detailed pseudocode implementation of this aggregation procedure is provided in Appendix B, Algorithm 1 for reproducibility.

### 3.4 MODEL ARCHITECTURES

We explore three families of state-of-the-art models: **Efficient CNNs:** EfficientNetV2 Tan & Le (2021) and ConvNeXt V2 Woo et al. (2023) serve as optimized convolutional baselines, well suited for $32 \times 32$ event images. **Vision Transformers:** ViT-CoMer Xia et al. (2024) and Swin Transformer V2 Liu et al. (2022) represent attention-based designs, capturing both local and global collision patterns. **State Space Models:** Mamba Gu & Dao (2023) and Vision Mamba Zhu et al. (2024) explore efficient alternatives to self-attention for long-range dependency modeling.

**Initialization and Input Resizing.** When training backbones, we consider multiple initialization strategies. Unless otherwise noted, models are trained from scratch with standard Gaussian initialization. For architectures providing a native $32 \times 32$ backbone variant (e.g., "Tiny"), we adopt it to reduce the interpolation overhead. Otherwise, input images are bilinearly upsampled from $32 \times 32$ to $224 \times 224$ before entering the backbone. The notation in Table 2 (e.g., "Tiny, Gaussian") indicates the combination of backbone size and initialization scheme used.

Table 2: Per-task performance of different backbones on the ANONYMIZED dataset. We report Accuracy and Macro-F1 for each physics parameter: Energy loss module, $\alpha_s$, and $Q_0$. All experiments use RLRP scheduler.

| Model | LR | Batch Size | Energy Loss | | $\alpha_s$ | | $Q_0$ | | $Acc_{total}$ (%) |
|---|---|---|---|---|---|---|---|---|---|
| | | | Acc (%) | F1 (%) | Acc (%) | F1 (%) | Acc (%) | F1 (%) | |
| EfficientNetV2 | $10^{-2}$ | 32 | 100.00 | 100.00 | 94.72 | 94.71 | 70.21 | 68.79 | 64.93 |
| ConvNeXt V2 (Tiny, Gaussian) | $10^{-4}$ | 32 | 100.00 | 100.00 | 93.54 | 93.53 | 74.10 | 73.59 | 67.64 |
| **ViT-CoMer (Tiny, Gaussian)** | $\mathbf{10^{-4}}$ | **32** | **100.00** | **100.00** | **95.83** | **95.83** | **78.19** | **77.57** | **74.50** |
| Swin Transformer V2 | $10^{-4}$ | 32 | 99.79 | 99.72 | 88.19 | 88.10 | 63.06 | 61.96 | 51.39 |
| Mamba (No Init) | $10^{-4}$ | 16 | 100.00 | 100.00 | 93.89 | 93.90 | 75.21 | 74.98 | 69.10 |

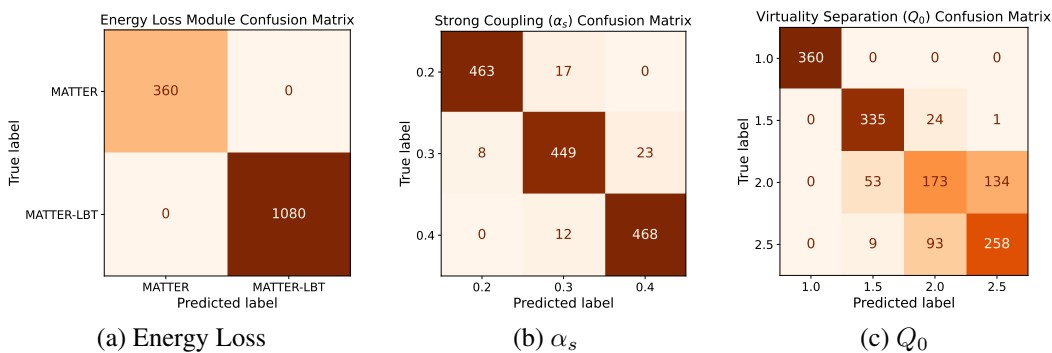

(a) Energy Loss  (b) $\alpha_s$  (c) $Q_0$

Figure 3: Confusion matrices for the best-performing models on each task. Energy loss classification is nearly perfect with no off-diagonal errors. For $\alpha_s$, mistakes are limited to adjacent classes ($0.2 \leftrightarrow 0.3$, $0.3 \leftrightarrow 0.4$), while for $Q_0$, confusion concentrates between $2.0$ and $2.5$. These structured errors indicate that misclassifications follow expected physics continuity rather than random noise.

## 4 EXPERIMENTS

We empirically measure the performance of state-of-the-art vision backbones on the ANONYMIZED dataset, following the training and evaluation protocol in Section 3. The experiments are designed to (i) establish baseline performance across model families, (ii) quantify the effect of loss weighting and optimization choices, and (iii) provide physics-informed analysis of prediction behavior.

**Training and Evaluation Protocol.** All models are trained on NVIDIA V100/A100 GPUs (16 GB, 1 CPU core) using the Adam optimizer with early stopping. We sweep learning rates $\{10^{-1}, 10^{-2}, 10^{-3}, 10^{-4}\}$ and batch sizes $\{16, 32, 64, 128, 256, 512\}$, reporting the best configuration by validation loss. Unless noted, we use the *RLRP scheduler (mode=max, factor=0.5, patience=4)*, with early stopping after three learning-rate reductions. The composite loss applies *Binary Cross-Entropy (BCE)* for $E$ and *Categorical Cross-Entropy (CCE)* for $\alpha_s$ and $Q_0$ with task weights $(\lambda_{\text{energy}}, \lambda_{\alpha_s}, \lambda_{Q_0})$. Evaluation includes per-task Accuracy, Macro-F1, Precision/Recall, and **joint accuracy** (exact match on $(E, \alpha_s, Q_0)$); confusion matrices analyze errors. Initialization choices (None, Gaussian, Tiny backbone) are detailed in Appendix E. All models use identical splits, normalization, and aggregation for fairness, with runtime statistics reported under consistent hardware.

### 4.1 BASELINES

We benchmark three families of models representing distinct inductive biases: **CNNs:** EfficientNetV2 and ConvNeXt V2, serving as convolutional baselines optimized for low-resolution inputs. **Vision Transformers:** ViT-CoMer and Swin Transformer V2, providing global context modeling through self-attention. **State Space Models:** Mamba, exploring linear-time sequence modeling for jet event representations.

A summary of model sizes, floating point operations (FLOPs), and average training time is provided in Table 1, highlighting efficiency trade-offs across architectures.

Table 3: Loss weight ablation for the composite objective. All experiments use the same `RLRP` scheduler, learning rate, and batch size mentioned in Table 2. We vary the task weights $(\lambda_{\text{energy}}, \lambda_{\alpha_s}, \lambda_{Q_0})$ (see Table 4 in the Appendix) and report per-task Macro-F1 as well as $Acc_{total}$.

| Model | Loss Weights | | | Energy Loss | | $\alpha_s$ | | $Q_0$ | | $Acc_{total}$ (%) |
|---|---|---|---|---|---|---|---|---|---|---|
| | $\lambda_{\text{energy}}$ | $\lambda_{\alpha_s}$ | $\lambda_{Q_0}$ | Acc (%) | F1 (%) | Acc (%) | F1 (%) | Acc (%) | F1 (%) | |
| **EfficientNetV2** | **0.6** | **1.6** | **0.8** | **100.0** | **100.0** | **95.5** | **95.5** | **78.5** | **78.7** | **74.0** |
| ConvNeXt V2 | 0.4 | 0.6 | 2.0 | 100.0 | 100.0 | 93.8 | 93.7 | 75.3 | 75.1 | 69.1 |
| ViT-CoMer | 0.8 | 0.8 | 1.4 | 100.0 | 100.0 | 96.2 | 96.2 | 76.7 | 76.9 | 72.9 |
| Swin Transformer V2 | 0.8 | 1.2 | 1.0 | 100.0 | 100.0 | 89.6 | 89.6 | 67.7 | 67.6 | 57.3 |
| Mamba | 0.6 | 0.8 | 1.6 | 100.0 | 100.0 | 94.4 | 94.5 | 74.7 | 75.0 | 69.1 |

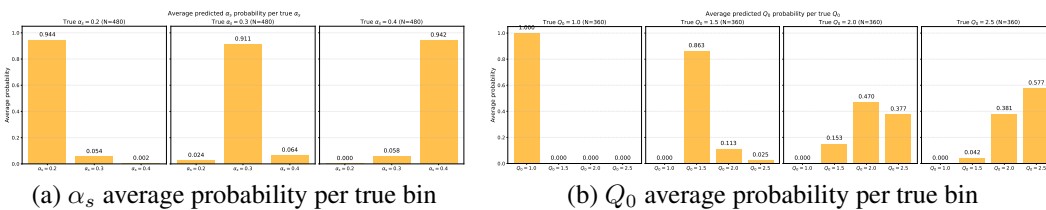

(a) $\alpha_s$ average probability per true bin      (b) $Q_0$ average probability per true bin

Figure 4: Average predicted probabilities per true bin for $\alpha_s$ (left) and $Q_0$ (right). Predictions are sharply peaked at the correct class, with probability mass leaking smoothly into neighboring bins. This structured calibration confirms that model uncertainties align with the expected continuity of QCD parameters, rather than arising from random noise.

## 4.2 ABLATIONS

To probe the robustness of our findings, we conduct controlled ablations:

**Learning schedule.** We compared fixed learning rate, step decay, cosine annealing, and `ReduceLROnPlateau` (RLRP). RLRP consistently provided the most stable convergence and highest accuracy across all backbones, so we report only RLRP results in Tables 2 and 3.

**Learning rate and batch size.** Grid search over LR × BS combinations, with the best setting per model reported in Table 2.

**Loss weighting.** We sweep over the task-weighting schemes (S1–S10, defined in Appendix D, Table 4) to probe how different emphasis on $Q_0$, $\alpha_s$, or Energy affect performance (Table 3).

## 5 RESULTS AND DISCUSSION

We now present results of JetBench, the multi-parameter classification using the ANONYMIZED dataset. Evaluation metrics serve both as computer vision benchmarks and physics consistency checks, as model errors align with known QGP dynamics.

**Overall Performance.** Table 2 reports the best per-task Accuracy and Macro-F1 for each backbone under the RLRP scheduler, using standard task weights $(1, 1, 1)$. Energy loss classification is saturated ($\approx 100\%$) across all families, while $\alpha_s$ and $Q_0$ remain more challenging. The strongest joint accuracy is achieved by ViT-CoMer ($74.5\%$). Mamba ($69.1\%$) and ConvNeXt ($67.6\%$) follow as competitive mid-tier performers, while EfficientNetV2 lags behind at $64.9\%$, and Swin Transformer underperforms at $51.4\%$.

Precision and Recall follow the same overall trends as Accuracy and F1: $Q_0$ consistently exhibits lower recall relative to precision, indicating that models tend to under-predict high-$Q_0$ bins. This systematic bias reflects the intrinsic difficulty of distinguishing high-$Q_0$ regimes rather than random misclassification noise; full Precision/Recall tables are provided in Appendix F (Tables 5).

**Loss Weighting Effects.** Table 3 summarizes ablations over task weights $(\lambda_{\text{energy}}, \lambda_{\alpha_s}, \lambda_{Q_0})$ (schemes S1–S10; see Table 4 in the Appendix D for definitions). Emphasizing $Q_0$ (S2–S5) consistently raises its Macro-F1 by up to 4 points but induces a small drop in $\alpha_s$, reflecting the inherent trade-off structure of the multi-task objective. Conversely, $\alpha_s$-strong weighting (S7) maximizes

| | $Acc_E$ | $F1_E$ | $Acc_{\alpha_s}$ | $F1_{\alpha_s}$ | $Acc_{Q_0}$ | $F1_{Q_0}$ | $Acc_{total}$ |
|---|---|---|---|---|---|---|---|
| ViT | 100.0 | 100.0 | 95.8 | 95.8 | 78.2 | 77.6 | 74.5 |
| EfficientNet | 100.0 | 100.0 | 95.5 | 95.5 | 78.5 | 78.7 | 74.0 |
| Mamba | 100.0 | 100.0 | 94.4 | 94.5 | 74.7 | 75.0 | 69.1 |
| ConvNeXt | 100.0 | 100.0 | 93.8 | 93.7 | 75.3 | 75.1 | 69.1 |
| Swin | 100.0 | 100.0 | 89.6 | 89.6 | 67.7 | 67.6 | 57.3 |

Figure 5: Heatmap of per-model metrics across tasks. Values show Accuracy and Macro-F1 for $E$, $\alpha_s$, $Q_0$, and joint accuracy.

joint accuracy (74.0% with EfficientNetV2), while $Q_0$-mid emphasis (S3) achieves the most balanced trade-off across tasks.

ViT-CoMer also benefits from $Q_0$-mild weighting (S2), and Swin Transformer improves modestly under $\alpha_s$-mild weighting (S6), though both follow the same trend. Detailed Precision/Recall values, provided in Appendix F, Table 6, confirm this pattern: stronger $Q_0$ weights improve recall on difficult bins at the expense of slight precision loss on $\alpha_s$. Overall, loss re-weighting provides a mechanism to target harder heads, but the optimal scheme depends on whether the goal is maximizing joint accuracy or prioritizing $Q_0$ fidelity.

**Backbone Comparisons.** Figure 5 consolidates per-model performance across all tasks, reporting Accuracy and Macro-F1 for Energy loss, $\alpha_s$, $Q_0$, and joint accuracy. All models saturate the binary energy-loss task ($\sim 100\%$), while $\alpha_s$ and especially $Q_0$ remain more challenging.

Among backbones, ViT-CoMer achieves the strongest joint accuracy (74.5%) in the baseline setting (S1), while EfficientNetV2 reaches a comparable peak (74.0%) under $\alpha_s$-strong weighting (S7). ConvNeXt V2 improves to 69.1% joint accuracy with $Q_0$-max emphasis (S5), and Mamba reaches 69.1% under $Q_0$-mid emphasis (S3). Swin Transformer V2 shows consistent underperformance, though it gains modestly from $\alpha_s$-mild weighting (S6), rising from $51.4\%$ to $57.3\%$.

Overall, Vision Transformers (ViT-CoMer) and Efficient CNNs (EfficientNetV2) deliver the strongest results, with ConvNeXt and Mamba competitive at mid-tier performance when task weights are tuned. Loss weighting generally improves $Q_0$ recall but does not alter the relative ranking: ViT-CoMer and EfficientNetV2 lead, followed by ConvNeXt and Mamba, with Swin trailing.

**Learning Dynamics.** Figure 6 illustrates convergence behavior. CNNs (e.g., ConvNeXt, EfficientNetV2) converge rapidly but plateau early. In particular, EfficientNetV2 briefly reaches $\sim 70$–$74\%$ joint accuracy, but its validation curves flatten quickly and early stopping is triggered after three learning-rate reductions, limiting further gains. Given these unstable learning curves, EfficientNetV2's reported accuracy should be interpreted with caution, as it reflects early saturation rather than sustained optimization.

In contrast, ViT-CoMer continues to improve throughout training under the RLRP scheduler, exhibiting smoother convergence and ultimately achieving the highest accuracy. Validation loss shows oscillations even as validation accuracy rises, reflecting imbalanced gradients across heads: the binary energy-loss task saturates, while the four-way $Q_0$ head remains unstable. Nevertheless, accuracy and Macro-F1 improve consistently, confirming that optimization progresses despite noisy losses. This contrast highlights distinct dynamics: CNNs saturate quickly, whereas Transformers retain capacity for sustained improvement. Additional curves for all backbones are provided in Appendix G, Figure 9.

**Structured Confusions and Calibration.** Figure 3 shows task-level confusion matrices for the best-performing ViT-CoMer model. Energy loss classification is essentially perfect, with no off-diagonal errors. For $\alpha_s$, misclassifications are restricted to adjacent bins ($0.2 \leftrightarrow 0.3$, $0.3 \leftrightarrow 0.4$), reflecting the smoothness of the underlying coupling strength. For $Q_0$, the main confusion occurs between 2.0 and 2.5, consistent with reduced discriminative signal at higher virtuality scales.

Calibration trends are shown in Figure 4, which plots average predicted probabilities per true bin for $\alpha_s$ and $Q_0$. Predictions are sharply peaked on the correct class, with probability mass smoothly

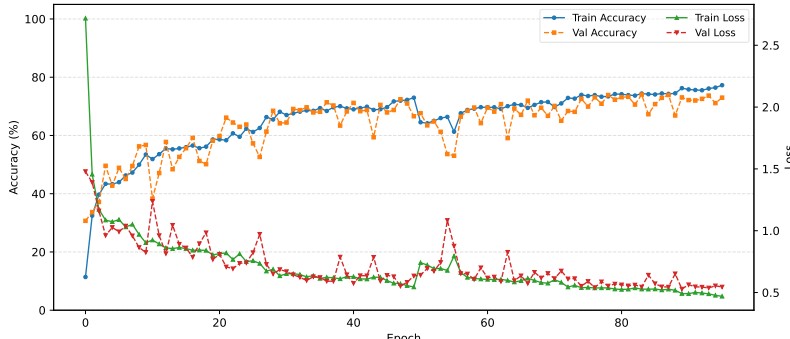

Figure 6: Learning curves (training/validation accuracy and loss) for ViT-CoMer. Validation accuracy improves steadily under the RLRP scheduler, while validation loss fluctuates due to the more difficult $Q_0$ head. Despite noisy optimization signals, the model achieves smooth and stable accuracy gains.

leaking into neighboring bins. This behavior corroborates the confusion matrices: errors respect the expected continuity structure of QCD parameters rather than appearing random. Together, these results demonstrate that the models are not only accurate but also calibrated in a physics-consistent manner, capturing the graded boundaries between adjacent parameter values.

**Discussion.** Key findings are: (1) Energy loss classification is saturated ($\sim$100%), while $\alpha_s$ (95–96%) and $Q_0$ (up to 78%) remain non-trivial; (2) Transformers, particularly ViT-CoMer, achieve the best joint accuracy ($\sim$74.5%), outperforming CNNs and Mamba by 5–7%, while Swin lags behind; (3) $Q_0$ consistently shows confusion between 2.0 and 2.5, reflecting limited discriminative signal in that regime; and (4) Loss re-weighting reveals clear trade-offs: stronger $Q_0$ emphasis improves its F1/recall but slightly reduces $\alpha_s$, while $\alpha_s$-strong weighting maximizes joint accuracy.

## 6 CONCLUSION AND FUTURE WORK

We present the first systematic study of **multi-parameter classification** of heavy-ion collisions with the ANONYMIZED dataset. Benchmarking CNNs, Vision Transformers, and state-space models shows the feasibility of jointly predicting the energy-loss module, $\alpha_s$, and $Q_0$ from $32 \times 32$ jet images. CNNs (EfficientNetV2, ConvNeXtV2) provide strong baselines but plateau early, while Transformers and Mamba variants better capture global context. ViT-CoMer with RLRP scheduling yields the most stable and accurate results. Performance saturates for energy loss, $\alpha_s$ remains moderately difficult, and $Q_0$ is hardest with errors concentrated between adjacent bins. Calibration analyses further show model uncertainties align with QCD continuity, underscoring deep models as scalable complements to Bayesian analysis.

**Future Work.** Promising directions include: (1) expanding the dataset to additional jet parameters; (2) diffusion-based generative models as fast surrogates for JETSCAPE-like simulations; (3) physics-informed architectures incorporating symmetries or conservation laws; (4) experimental validation on RHIC/LHC detector data; (5) adaptive or uncertainty-based loss weighting; (6) multi-moment jet representations beyond the first moment; and (7) richer evaluation metrics, including calibration curves and uncertainty quantification.

**Broader Impacts and Limitations.** This study shows the potential of ML in heavy-ion physics but remains limited by simulated data, hyperparameter sensitivity, and the compute cost of modern backbones. While ViT-CoMer and EfficientNetV2 offer strong baselines, validation on experimental data is needed for transferability. Because each image aggregates 500 events and the dataset is fully shuffled, run-to-run variance is statistically suppressed; small-scale trials further suggest low variance. We therefore report single-seed results, with multi-seed robustness to be added in the camera-ready version. Nonetheless, our results chart a scalable path toward physics-informed deep learning, complementing Bayesian inference while underscoring the need for experimental validation.

## REPRODUCIBILITY STATEMENT

We have taken several steps to ensure the reproducibility of our results. The ANONYMIZED dataset and preprocessing pipeline are described in Section 3.3, with pseudocode for the virtual image aggregation in Appendix B. Initialization strategies and training protocols, including hyperparameters and learning schedules, are detailed in Section 4 and Appendix E. Complete ablation results, precision/recall metrics, and calibration analyses are reported in the Appendix. All backbone results are reported from single-seed runs; because each image averages 500 events and the dataset is fully shuffled, the variance across runs is expected to be low, and preliminary small-scale trials support this observation. For completeness, multi-seed evaluations will be included in the camera-ready version. To further support reproducibility, we release the full training and evaluation code as anonymized supplementary material; upon acceptance, we will open-source this repository.

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

# APPENDIX

## A    RELATIVISTIC HEAVY-ION COLLISION SCHEMATIC

To complement the main text, we include a schematic overview of the stages of a relativistic heavy-ion collision. While the ANONYMIZED dataset focuses on final-state particle distributions represented as $32 \times 32$ images, this figure provides broader context by illustrating the physical processes that lead to these detector-level patterns. It highlights how two ultra-relativistic nuclei collide, form a short-lived quark–gluon plasma (QGP), and subsequently hadronize into final-state particles. These final distributions are the basis of the images used in our classification benchmark (see Fig. 1 in the main paper).

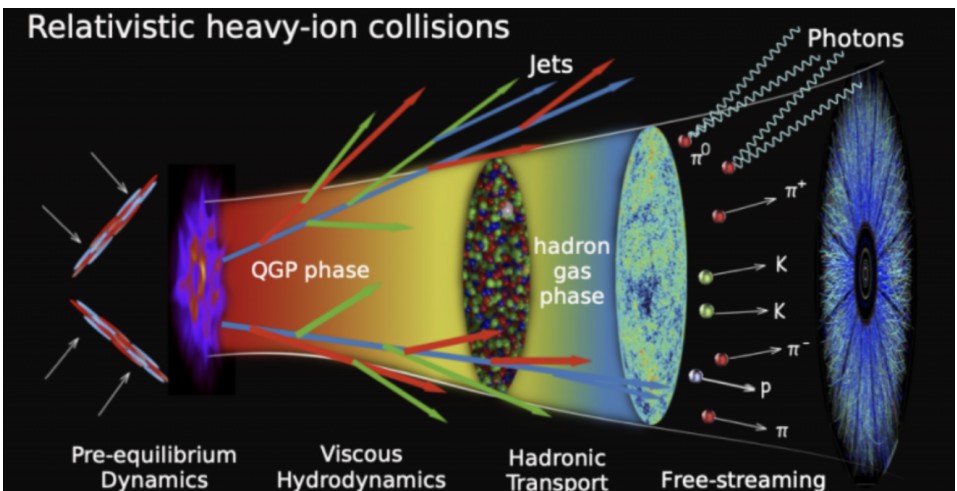

Figure 7: Schematic overview of relativistic heavy-ion collisions. Two heavy nuclei (e.g., Pb–Pb at the LHC or Au–Au at RHIC) collide at ultra-relativistic energies, creating extreme temperature and density conditions. The collision produces a quark–gluon plasma (QGP), which expands, cools, and hadronizes into final-state particles. These particles are detected as jets and soft hadrons, forming the inputs to the ANONYMIZED dataset ANONYMIZED (2025).

## B    VIRTUAL IMAGE AGGREGATION ALGORITHM

For reproducibility, we provide pseudocode for the proposed momentum-based aggregation method. This algorithm groups raw event images with identical $(\alpha_s, Q_0, \text{module})$ labels and averages them into a single "virtual" profile, producing a CSV index of aggregated samples. This implementation underlies the moment-based aggregation method described in Section 3.3.

## C    VISUALIZATION OF AGGREGATED EVENT SAMPLES

To illustrate the variability of the ANONYMIZED dataset, we visualize aggregated two-dimensional histograms of Pb–Pb collision events in the pseudorapidity–azimuthal plane $(\eta, \phi)$, constructed using the momentum-based aggregation procedure introduced in Section 3.3. Figure 8 shows a $12 \times 10$ grid of averaged samples, where each row corresponds to a distinct combination of the energy loss module (MATTER or MATTER–LBT), strong coupling constant $\alpha_s \in \{0.2, 0.3, 0.4\}$, and virtuality separation scale $Q_0 \in \{1.0, 1.5, 2.0, 2.5\}$. For each $(\alpha_s, Q_0, E)$ setting, ten representative aggregated events are displayed. The color scale encodes normalized transverse momentum $p_T$, highlighting the jet energy deposition pattern across different parameter regimes. This visualization demonstrates the diversity of event structures across the design space. Differences in intensity and spread of the distributions reflect the underlying parton energy loss mechanisms and medium response. These aggregated samples provide intuition for the learning task: predicting the physics

---

**Algorithm 1** Virtual Image Aggregation for Dataset Construction

---

**Require:** Dataset root directory $\mathcal{D}$ with labeled event images as .npy files
**Require:** Label parser `parse_labels(dir)` $\rightarrow (e, \alpha, Q_0)$
**Require:** Aggregation group size $k$
**Ensure:** Aggregated file-label CSV with $N_{\text{agg}}$ entries

 1: Initialize $\mathcal{G} \leftarrow \emptyset$                          ▷ Dictionary mapping label tuples to file paths
 2: **for all** directory $d \in$ `os.listdir`$(\mathcal{D})$ **do**
 3:     $label \leftarrow$ `parse_labels`$(d)$
 4:     **for all** file $f \in$ `os.listdir`$(d)$ such that $f$ ends with '.npy' **do**
 5:        $\mathcal{G}[label]$.append$(f)$
 6:     **end for**
 7: **end for**
 8: Initialize empty list $\mathcal{A}$ for aggregated samples
 9: $id \leftarrow 0$
10: **for all** $label \in$ `keys`$(\mathcal{G})$ **do**
11:     Shuffle $\mathcal{G}[label]$
12:     **for** $i = 0$ to $|\mathcal{G}[label]| - k$ step $k$ **do**
13:        $group \leftarrow \mathcal{G}[label][i : i + k]$
14:        $agg\_id \leftarrow$ "agg_" $\|$ `zfill`$(id)$
15:        $\mathcal{A}$.append$((agg\_id, group, label))$
16:        $id \leftarrow id + 1$
17:     **end for**
18: **end for**
19: Save $\mathcal{A}$ as a CSV file with columns: `agg_id`, `file_paths`, $e, \alpha, Q_0$

---

parameters $(E, \alpha_s, Q_0)$ directly from sparse event images is non-trivial, yet crucial for advancing data-driven modeling in heavy-ion physics.

## D   Task-Weighting Schemes

In Section 3, we introduced the composite objective

$$\mathcal{L}_{\text{total}} = \lambda_{\text{energy}} \mathcal{L}_{\text{energy}} + \lambda_{\alpha_s} \mathcal{L}_{\alpha_s} + \lambda_{Q_0} \mathcal{L}_{Q_0}.$$

To explore how emphasizing different heads influences performance, we designed a set of interpretable weighting schemes. Each scheme is identified by a shorthand name (S1–S10), the triplet of coefficients $(\lambda_{\text{energy}}, \lambda_{\alpha_s}, \lambda_{Q_0})$, and a short rationale.

Table 4: Loss-weighting schemes used in our ablations.

| ID | $\lambda_{\text{energy}}$ | $\lambda_{\alpha_s}$ | $\lambda_{Q_0}$ | Description |
|----|------|------|------|-------------|
| S1 | 1.0 | 1.0 | 1.0 | Balanced (equal weights) |
| S2 | 0.8 | 0.8 | 1.4 | $Q_0$ mild emphasis |
| S3 | 0.6 | 0.8 | 1.6 | $Q_0$ mid emphasis |
| S4 | 0.5 | 0.7 | 1.8 | $Q_0$ strong emphasis |
| S5 | 0.4 | 0.6 | 2.0 | $Q_0$ max emphasis |
| S6 | 0.8 | 1.2 | 1.0 | $\alpha_s$ mild emphasis |
| S7 | 0.6 | 1.6 | 0.8 | $\alpha_s$ strong emphasis |
| S8 | 1.2 | 0.9 | 0.9 | Energy bump (binary head stress test) |
| S9 | 0.9 | 1.0 | 1.1 | $Q_0$ very mild emphasis |
| S10 | 0.7 | 1.0 | 1.3 | $Q_0 + \alpha_s$ paired emphasis |

These schemes provide structured ablation points that make it possible to compare trends across backbones. The full sweep results for each scheme are reported in Table 3 of the main paper.

## E   Initialization Strategies

We evaluated three initialization settings during backbone training:

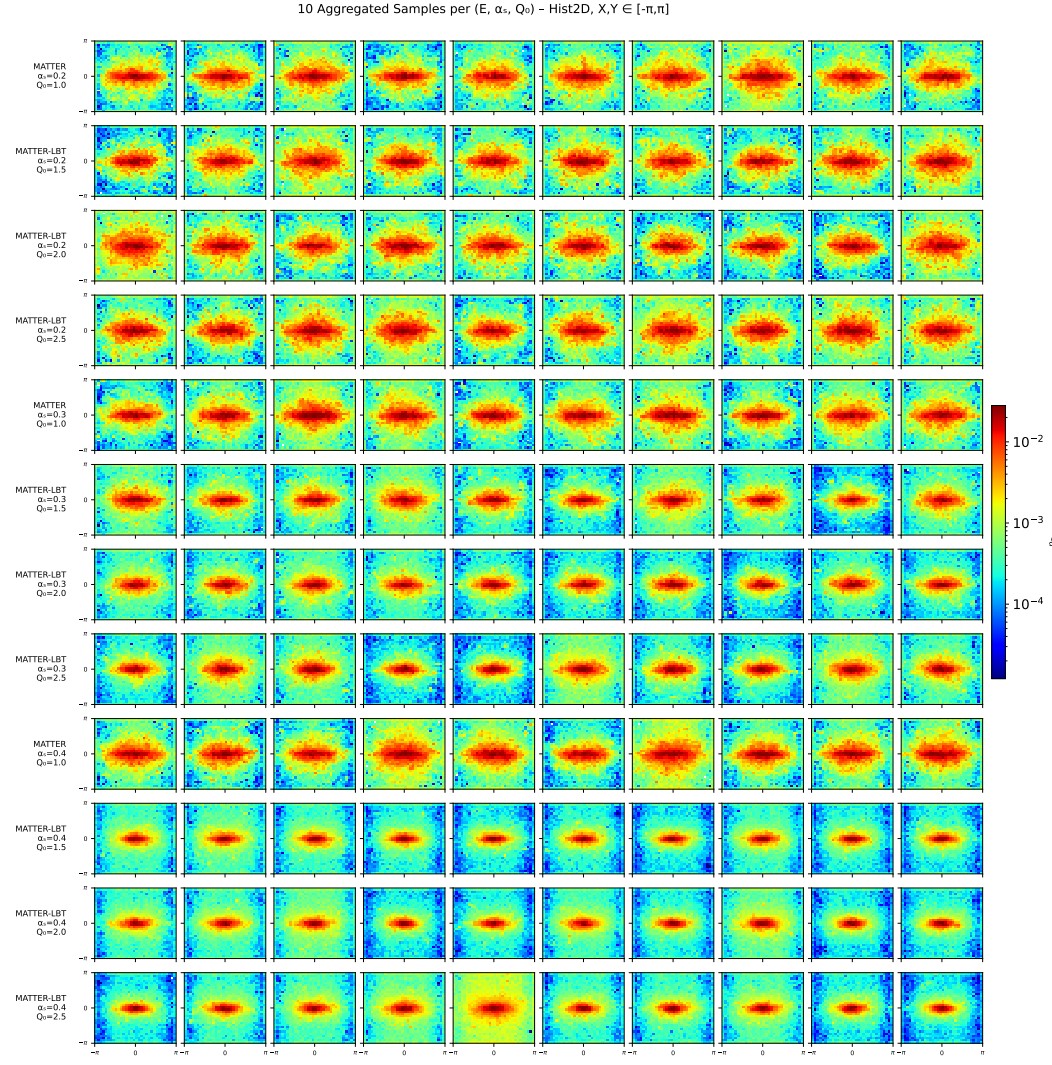

Figure 8: Grid of aggregated Pb–Pb collision events from the ANONYMIZED dataset. Each row corresponds to a unique combination of energy loss module (MATTER or MATTER–LBT), strong coupling constant $\alpha_s \in \{0.2, 0.3, 0.4\}$, and virtuality separation scale $Q_0 \in \{1.0, 1.5, 2.0, 2.5\}$. For each $(E, \alpha_s, Q_0)$ setting, ten representative aggregated event images are shown as 2D histograms in the $(\eta, \phi)$ plane, with color intensity indicating normalized transverse momentum $p_T$.

(i) **None:** training from scratch with framework default weight initialization. For fully connected or convolutional layers, these are typically variance-preserving schemes such as Xavier Glorot & Bengio (2010) or Kaiming He et al. (2015), designed so that the variance of activations remains stable across layers:

$$\text{Var}[W_{ij}] = \begin{cases} \frac{2}{n_{\text{in}}+n_{\text{out}}}, & \text{(Xavier)} \\ \frac{2}{n_{\text{in}}}, & \text{(Kaiming ReLU)}, \end{cases} \tag{7}$$

where $n_{\text{in}}$ and $n_{\text{out}}$ are the input and output dimensions.

(ii) **Gaussian:** a custom scheme in which each weight matrix $W$ of a linear or convolutional layer is sampled i.i.d. from a zero-mean Gaussian:

$$W_{ij} \sim \mathcal{N}(0, \sigma^2), \quad \sigma = 0.02, \tag{8}$$

with all bias terms initialized to zero. This corresponds to the PyTorch routine:

$$W \leftarrow \texttt{Normal}(0, 0.02), \quad b \leftarrow 0, \tag{9}$$

Table 5: Per-task Precision and Recall for each backbone. Metrics are reported separately for Energy loss, $\alpha_s$, and $Q_0$. Accuracy and F1 are reported in the main paper (Table 2).

| Model | Energy Loss | | $\alpha_s$ | | $Q_0$ | |
|---|---|---|---|---|---|---|
| | Prec (%) | Rec (%) | Prec (%) | Rec (%) | Prec (%) | Rec (%) |
| EfficientNet V2 | 100.0 | 100.0 | 94.9 | 94.7 | 76.9 | 70.2 |
| ConvNeXt V2 (Tiny, Gaussian) | 100.0 | 100.0 | 93.5 | 93.5 | 73.4 | 74.1 |
| **ViT-CoMer (Tiny, Gaussian)** | **100.0** | **100.0** | **95.9** | **95.8** | **77.4** | **78.2** |
| Swin Transformer V2 | 99.6 | 99.9 | 88.6 | 88.2 | 61.8 | 63.1 |
| Mamba | 100.0 | 100.0 | 93.9 | 93.9 | 75.1 | 75.2 |

Table 6: Per-task Precision and Recall for the loss weight ablation study. All experiments use the same scheduler (RLRP), learning rate, and batch size as in Table 3. We vary the task weights $(\lambda_{\text{energy}}, \lambda_{\alpha_s}, \lambda_{Q_0})$ (schemes S2–S7).

| Model | Loss Weights | | | Energy Loss | | $\alpha_s$ | | $Q_0$ | |
|---|---|---|---|---|---|---|---|---|---|
| | $\lambda_{\text{energy}}$ | $\lambda_{\alpha_s}$ | $\lambda_{Q_0}$ | Prec (%) | Rec (%) | Prec (%) | Rec (%) | Prec (%) | Rec (%) |
| **EfficientNet V2** | **0.6** | **1.6** | **0.8** | **100.0** | **100.0** | **95.8** | **95.5** | **79.7** | **78.5** |
| ConvNeXt V2 | 0.4 | 0.6 | 2.0 | 100.0 | 100.0 | 93.8 | 93.8 | 75.2 | 75.3 |
| ViT-CoMer | 0.8 | 0.8 | 1.4 | 100.0 | 100.0 | 96.2 | 96.2 | 77.2 | 76.7 |
| Swin Transformer V2 | 0.8 | 1.2 | 1.0 | 100.0 | 100.0 | 89.6 | 89.6 | 67.5 | 67.7 |
| Mamba | 0.6 | 0.8 | 1.6 | 100.0 | 100.0 | 94.5 | 94.4 | 75.9 | 74.7 |

applied layer-wise. Such initialization stabilizes gradients in the early epochs by bounding the variance of activations.

(iii) **Tiny backbone:** when available (e.g., ViT-Tiny), we trained the $32 \times 32$ variant directly to avoid interpolation overhead. For models without native $32 \times 32$ support, input images were upsampled to $224 \times 224$ using bilinear interpolation:

$$I_{224}(x,y) = \sum_{i,j} I_{32}(i,j)\, \phi(x/7 - i)\, \phi(y/7 - j), \tag{10}$$

where $\phi(\cdot)$ is the bilinear kernel.

These design choices are reflected in Table 2.

## F  ADDITIONAL METRICS: PRECISION AND RECALL

For completeness, we report per-task Precision and Recall values in addition to Accuracy and Macro-F1. Table 5 presents results for all backbones, while Table 6 extends this analysis to the loss weight ablation study. Across both backbones and weighting schemes, the results corroborate our main findings: Precision and Recall follow the same trends as Accuracy and F1, with $Q_0$ consistently emerging as the most challenging head.

## G  LEARNING CURVES

To complement the main text (Section 5), we provide complete learning curves for all backbones. Each figure shows training and validation accuracy as a function of epochs under the `ReduceLROnPlateau` scheduler. These curves illustrate convergence dynamics across architectures: CNNs converge quickly but plateau early, Transformers benefit from smoother training, and Mamba exhibits slower yet steady convergence.

An additional observation is that EfficientNetV2 reaches a joint accuracy of $\sim 74\%$ but triggers early stopping once validation accuracy plateaus, whereas ViT-CoMer continues to improve throughout training. This highlights differences in optimization dynamics: CNNs can saturate rapidly, while Transformers retain capacity for further gains with extended training.

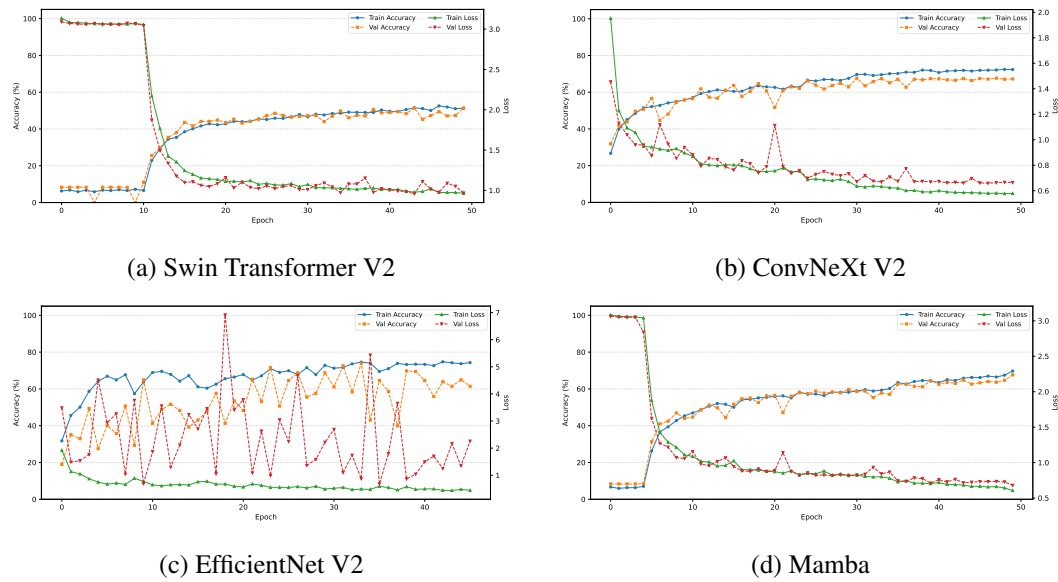

(a) Swin Transformer V2

(b) ConvNeXt V2

(c) EfficientNet V2

(d) Mamba

Figure 9: Learning curves (training/validation accuracy vs. epochs) for four backbones on ANONYMIZED under `ReduceLROnPlateau`. CNNs (EfficientNetV2, ConvNeXt) converge quickly but plateau earlier; ViT-CoMer shows smoother improvements; Mamba converges more slowly yet steadily.

## H  USE OF LLMs

We used ChatGPT as a general-purpose assistive tool in limited parts of this work. Specifically, ChatGPT was employed to:

- **Polish writing**, including improving clarity, conciseness, and readability of the manuscript.
- **Reformat technical content**, such as adjusting figure/table captions and ensuring consistent referencing style.
- **Assist with anonymization**, helping to identify and rephrase text that could reveal author identity.

ChatGPT was not involved in research ideation, dataset design, experimental setup, or analysis. All scientific ideas, methods, experiments, and conclusions were developed independently by the authors.

