# OpenReview forum: "JetBench: Benchmarking Vision Models for Jet Observables' Classification in Heavy-Ion Physics"
_ICLR.cc/2026/Conference — Submitted to ICLR 2026_

### Official Review · Reviewer_3BQD · 2025-11-01

**Soundness:** 3
**Presentation:** 3
**Contribution:** 2
**Rating:** 4
**Confidence:** 3

**Summary:**

This paper introduces JetBench, a benchmark for multi-parameter classification of averaged jet images in heavy-ion physics. Each event is a $32\times32$ $(\eta, \phi)$ jet image with labels for energy-loss module (MATTER vs MATTER–LBT), $\alpha_s \in \{0.2,0.3,0.4\}$, and $Q_0 \in \{1.0,1.5,2.0,2.5\}$. To stabilize learning on sparse events, the authors aggregate many events sharing the same labels into a mean jet image. They train CNNs, Vision Transformers, and state-space models with a three-component loss function and evaluate per-task metrics and joint exact-match. Results show near-perfect energy-loss, $\sim$95% $\alpha_s$, and up to 78% $Q_0$, with ViT-CoMer achieving 74.5% joint accuracy. Confusion is structured (e.g., α_s errors are adjacent bins; $Q_0$ mainly 2.0 vs 2.5), suggesting the models are learning the underlying physics.

**Strengths:**

* The paper proposes a novel, expanded benchmark task of multi-parameter estimation ($\alpha_s$, $Q_0$, and energy loss module).
* The moment-based aggregation of the jet images is an interesting technique to stabilize training.
* Comprehensive side-by-side comparisons of CNN, ViT, and SSM backbones with a unified training protocol (common splits, schedulers, loss weighting).
* Patterns of confusion concentrated in adjacent bins suggest that the models are learning the underlying physics, with Q_0 as the most difficult parameter to learn.
* Provides standardized multi-parameter baselines for future comparisons that can motivate future work.

**Weaknesses:**

* The paper adopts a three-parameter objective, but does not motivate this design or compare to training three independent models. Another alternative is training an encoder in a self-supervised way and tuning the classification head for different tasks.
* The label-conditioned averaging is only feasible in simulation. For data, labels are unknown and event sparsity remains. Consider (i) training on small-N aggregates and testing on single events, (ii) self-supervised pretraining on single events followed by fine-tuning on aggregates
* Only accuracy/F1 are reported. Please include threshold-independent metrics (AUROC/AUPRC) as well.
* Even if variance is expected to be small, main tables should include multi-seed means ± std (or CIs) and, for the best model, learning-curve variability across seeds.
* Given per-event sparsity and the physics of jets, set/point-cloud models (PFN/EFN/ParticleNet/ParT, transformers with permutation invariance) are natural baselines and may be more sensitive to Q_0. At a minimum, consider including one strong particle-cloud baseline and discuss inductive biases vs. images.
* The curated, balanced label grid likely diverges from real class priors and experimental conditions. Add experiments on (i) prior shift (train balanced, test imbalanced), (ii) centrality/background variations, and (iii) detector effects (smearing, pileup).

**Questions:**

- What is the empirical benefit of joint three-head training over (a) three independent models and (b) shared-trunk + per-task heads with/without uncertainty-based weighting? Please provide a comparison.
- Since label-conditioned averaging is not available for data, how do the authors envision applying the moment-based aggregation framework? Have the authors evaluated training with smaller aggregation $N$ and testing on single events (or weakly-supervised/EM-style aggregation)? In addition, how are jet images centered for the purpose of aggregating?
- Given event sparsity, did the authors evaluate point-cloud / set-based architectures or hybrid “image + set” models?
- The dataset is balanced and enforces specific (module, $Q_0$) validity rules. How does performance change under realistic (imbalanced) priors and under domain shifts (centrality, background, detector effects)?
- Can the authors add AUROC/AUPRC for each head? These would clarify ranking ability and uncertainty quality beyond accuracy/F1.
- Please provide multi-seed results (mean $\pm$ std) for main tables and learning-curve variability for the best model to confirm robustness of conclusions.

---

### Official Review · Reviewer_LNz6 · 2025-11-01

**Soundness:** 3
**Presentation:** 3
**Contribution:** 2
**Rating:** 2
**Confidence:** 3

**Summary:**

The paper benchmarks some standard CNN-based architectures (including vision transformers) on a single particle jet tagging dataset.

**Strengths:**

The paper presents a fair evaluation of different state of the art architectures on a problem of practical relevance to the hep community.

**Weaknesses:**

The paper's contribution is quite incremental. No new algorithms or architectures are presented. Earlier work by the (anonymized) same authors evaluates similar algorithms on the same dataset, with the only apparent difference being that the earlier work concentrated on just on predicting energy loss, whereas this paper also predicts two additional quantities.

**Questions:**

It is a bit difficult to evaluate the paper because both the dataset and the prior work benchmarking algorithms with similar architectures but only for energy loss are cited as "anonymized". Is there any other difference between this paper and your prior work besides the fact that you now have 3 target quantities?

---

### Official Review · Reviewer_5ZrH · 2025-11-01

**Soundness:** 3
**Presentation:** 3
**Contribution:** 1
**Rating:** 2
**Confidence:** 3

**Summary:**

I am by no means a physics expert. I tried my best to review this paper from the viewpoint of ICLR.

This paper presents JetBench, a benchmark for multi-parameter classification of relativistic heavy-ion collision events. Each event is represented as a 32×32 jet image, labeled with three physics parameters: (i) Energy loss module, (ii) Strong coupling constant, and (iii) Virtuality separation scale (Q0).

The dataset, built from JETSCAPE simulations, contains 7.2 million aggregated jet-event images. The authors benchmark several modern vision architectures, namely EfficientNetV2, ConvNeXt V2, ViT-CoMer, Swin V2, and Mamba, for joint prediction of these parameters.  A key methodological component is the Virtual Image Aggregation technique, which averages multiple sparse jet events. The paper also analyzes calibration and confusion patterns, showing that model errors follow smooth, physically consistent transitions. ViT-CoMer achieves the best overall performance.

**Strengths:**

Comprehensive benchmarking: Systematic evaluation of multiple model families under unified settings.

Clear methodology and reproducibility: Dataset construction, pseudocode, and ablation details are well documented.

Cross-disciplinary value: Establishes a link between deep vision models and heavy-ion collision physics, potentially useful for domain researchers.

High-quality presentation: The paper is well written, figures are clear, and experiments are carefully organized.

**Weaknesses:**

Limited ML novelty: The work benchmarks existing architectures without introducing new algorithms, losses, or theoretical insights.

No statistical robustness: Single-seed results are reported without standard deviations or significance testing.

Domain specificity: The contribution’s relevance to the ICLR audience is narrow, as the study’s core advances lie in computational physics rather than machine learning.

The paper is a well-executed application and benchmarking study, but its contribution to ICLR topics is incremental. It will be valuable for the physics community as a dataset and benchmarking resource, yet it lacks methodological or conceptual innovation aligned with ICLR’s core scope.

**Questions:**

I do not really have any specific questions. The authors could try to convince us about why this paper is relevant for the ICLR community.

---

### Official Review · Reviewer_Wsr2 · 2025-11-03

**Soundness:** 1
**Presentation:** 2
**Contribution:** 1
**Rating:** 2
**Confidence:** 4

**Summary:**

The authors perform multi-parameter classification using an ANONYMIZED dataset, which was previously published by the authors in a separate venue. They benchmark several neural network architectures on this dataset, including CNNs (EfficientNetV2, ConvNeXt V2), Vision Transformers (ViT-CoMer, Swin V2), and state space models (Mamba).

**Strengths:**

The study is supported by a comprehensive set of experiments, and the authors' transparency regarding their use of LLMs is commendable.

**Weaknesses:**

The technical approach lacks sufficient novelty for a top-tier venue like NeurIPS. The paper applies a standard set of existing model architectures (CNNs, Vision Transformers, Mamba) without introducing a novel methodological contribution, architectural innovation, or new theoretical insight. To meet the bar for NeurIPS, the work would need to go beyond a well-executed benchmark and present a significant conceptual advance.

**Questions:**

- Could the authors provide a more self-contained description of the dataset? Understanding the data generation methodology and the metadata is crucial for assessing the experimental setup and the generalizability of the results.
- I encourage the authors to consider whether a regression formulation might be more suitable for this task. Predicting continuous parameters via classification inherently introduces discretization error. A discussion of this point, or empirical results comparing against a regression baseline, would significantly strengthen the methodological rationale.

---

### Meta-Review · Area_Chair_sTfK · 2026-01-06

**Summary:**

The main concerns raised by the reviewers are:

1. The paper applies a standard set of existing model architectures (CNNs, Vision Transformers, Mamba) without introducing a novel methodological contribution, architectural innovation, or new theoretical insight (by all reviewers)

2. No statistical robustness: Single-seed results are reported without standard deviations or significance testing (by reviewer 5ZrH).

3. Domain specificity: The contribution’s relevance to the ICLR audience is narrow, as the study’s core advances lie in computational physics rather than machine learning (by reviewer 5ZrH).

**Reviewer Concerns:**

The authors did not provide a rebuttal, so no concerns are addressed.

**Reviewer Scores:**

The reviewers' scores will remain the same.

---

### Decision · Program_Chairs · 2026-01-26

Reject